# Heat-Killed *Enterococcus faecalis* EF-2001 Induces Human Dermal Papilla Cell Proliferation and Hair Regrowth in C57BL/6 Mice

**DOI:** 10.3390/ijms23105413

**Published:** 2022-05-12

**Authors:** Young-Hyun Baek, Jin-Ho Lee, Sang-Jin Chang, Yuri Chae, Myung-Hun Lee, Sun-Hong Kim, Kwon-Il Han, Tack-Joong Kim

**Affiliations:** 1Division of Biological Science and Technology, Yonsei University, Wonju 26493, Korea; snfjddl123@naver.com (Y.-H.B.); drlogos@naver.com (J.-H.L.); csj_korea@naver.com (S.-J.C.); dbfl7636@naver.com (Y.C.); onlyonce05@naver.com (M.-H.L.); kihan@berm.co.kr (K.-I.H.); 2Research & Development Center, Doctor TJ Co., Ltd., Wonju 26493, Korea; auscrlt1028@gmail.com; 3Research & Development Center, Bereum Co., Ltd., Wonju 26361, Korea

**Keywords:** EF-2001, heat-killed *Enterococcus faecalis*, hair regrowth, human dermal papilla cells, minoxidil

## Abstract

Minoxidil is the most widely used treatment for hair growth, but has been associated with several side effects. In this study, we investigated the effects of heat-killed *Enterococcus faecalis* EF-2001 on hair loss prevention and regrowth using human dermal papilla cells and male C57BL/6 mice. To examine the effects of EF-2001, we used minoxidil as the positive control. In the in vitro experiments, EF-2001 treatment (75–500 μg/mL) led to the proliferation of human dermal papilla cells in a concentration-dependent manner. In the in vivo experiment, the topical application of 200 µL EF-2001 on the dorsal surface of C57BL/6 male mice led to hair growth. Changes in hair regrowth were examined by visual comparison and hematoxylin and eosin staining of skin sections. We also determined the expression levels of marker genes (*Wnt*) and growth factors (fibroblast growth factor, insulin growth factor 1, and vascular endothelial growth factor) in the skin tissues of the back of each mouse using a quantitative polymerase chain reaction. EF-2001 accelerated the progression of hair regrowth in mice and promoted hair-follicle conversion from telogen to anagen, likely by increasing the expression levels of growth factors and marker genes.

## 1. Introduction

Hair plays various roles in the human body, including protecting and maintaining the temperature of the head [1]. Hair loss occurs because of aging, but recently, various factors such as genetic factors, stress, dietary habits, nutritional imbalance, and social activities have been associated with hair loss [2,3].

Hair loss in the South Korean population has been steadily increasing, from approximately 180,000 cases in 2009 to approximately 210,000 cases in 2013, with an average annual growth rate of 3.6% in the last 5 years. Of late, both middle-aged men and young women have shown hair-loss symptoms [4,5].

Androgenetic alopecia, also known as male and female pattern baldness, is the most common type of hair loss in postpubertal men and women [6,7]. Furthermore, androgenetic alopecia in men has been associated with several other medical conditions, including coronary heart disease and prostate enlargement. In women, this form of hair loss is associated with an increased risk of polycystic ovary syndrome.

Hair formation follows a regular cycle. The hair-growth cycle comprises three stages: anagen, catagen, and telogen. The anagen phase involves the rapid proliferation of matrix cells in the hair bulb; catagen involves apoptosis, leading to regression of the lower two-thirds of the follicle and preserving the stem cell region; and telogen is a relatively inactive period between growth phases. Hair grows or falls out during these cycles. In particular, during the catagen phase, apoptosis of many cells around the hair follicle progresses, and the size of the hair follicle decreases as telogen starts [8,9]. Therefore, studies investigating hair formation and fall should also consider hair growth and related factors [vascular endothelial growth factor (VEGF), fibroblast growth factor (FGF), insulin growth factor (IGF) 1, and Wnt ligands) that influence the hair cycle. In dermal papilla (DP) cells, FGF7 and FGF10 play a role in maintaining the anagen phase for long periods. FGF2 plays a role in increasing the proliferation of DP cells, thereby modulating FGF7 and FGF10 expression. Wnt signaling in DP cells plays a vital role in regulating hair-follicle morphogenesis, differentiation, and follicle regeneration. Therefore, it is a major factor enabling new hair to grow due to hair-follicle conversion from telogen to anagen. VEGF promotes angiogenesis around hair follicles and induces hair anagen. IGF1 is an important growth factor that regulates hair growth; it is produced only in mesenchymal cells, and its receptors are present in hair-follicle cells. Therefore, secreted IGF1 promotes the proliferation of hair-follicle cells and significantly increases the length of hair-follicle tissues [10,11,12,13,14].

Many products and medicines to overcome hair loss are currently available, and others are being developed [15]. For example, minoxidil has been approved by the United States Food and Drug Administration (FDA) to treat alopecia. Minoxidil was initially developed as a systemic antihypertensive agent [16]. In addition to its originally intended effects, it showed hair-growth stimulation as a side effect and is thus used to treat hair loss [17]. Several studies have reported that minoxidil promotes nutrient supply through vasodilation around hair follicles, such as through potassium channel opening [9,18,19]; induces hair growth; and promotes DP cell growth [20]. However, minoxidil provides a short-term improvement, and discontinuing treatment may result in rapid hair loss [17,21]. In addition, several side effects of minoxidil have been reported in clinical cases (such as minoxidil-mediated weight gain, edema, increased heart rate, angina, dermatitis, and itching), considerably restricting hair-loss treatment [22]. Therefore, several studies have focused on preventing hair loss and developing substances (derived from natural materials) with distinct hair-growth effects and few side effects [23,24,25,26].

*Enterococcus faecalis*, a Gram-positive commensal microorganism in the intestines, has immunostimulatory or regulatory effects [27,28]. Heat-killed *E. faecalis* has previously been shown to have beneficial effects on human health, including muscle atrophy prevention [29] and anti-inflammation [30]. Moreover, heat-killed *E. faecalis* derived from the intestines has been reported to soothe the skin and improve atopic disease in mice [31,32]. Recently, Fan et al. and Lee et al. reported on the biological effect and mechanism of heat-killed parabiotic *E. faecalis* on high-fat diet-induced obese rats and adipogenesis in 3T3-L1 cells [33,34].

However, little is known about the role of heat-killed *E. faecalis* in hair regrowth. Therefore, in this study, we analyzed the effect of EF-2011 on hair growth in human dermal papilla cells (HDPCs) and a C57BL/6 mouse model, and its potential as a hair-growth treatment was evaluated in comparison with minoxidil.

## 2. Results

### 2.1. Effect of EF-2001 on HDPC Proliferation

HDPCs play an important role in regulating hair growth. Therefore, we examined the effects of EF-2001 on HDPC proliferation using the MTT assay. The proliferation rate of HDPCs increased significantly after treatment with EF-2001 at 50, 75, 150, and 250 µg/mL for 48 h. At 150–500 μg/mL EF-2001, HDPC proliferation increased 1.5-fold (Figure 1).

### 2.2. EF-2001 Induced the Anagen Stage in C57BL/6 Mice 

We determined the effect of EF-2001 treatment on hair regrowth in C57BL/6 mice. Since the dorsal hair of these mice underwent a time-synchronized growth cycle, they were shaved 2 d before treatment. We compared the in vivo hair regrowth results of EF-2001 treatment with those of the untreated control group and the 5% minoxidil-treated group, which we considered a positive standard for hair-regrowth treatment. After 14 d, hair in the minoxidil-treated mice filled most of the dorsal area, whereas in control mice, a significantly lower rate of hair regrowth and pink shaved skin were observed in the dorsal area. The EFL (EF-2001 low concentration) and EFH (EF-2001 high concentration) groups showed results similar to those of the minoxidil-treated group. However, hair growth in the EFC group (3:1 mixture of acetone and olive-oil-treated group) was more than that in the control (PBS-treated group) (Figure 2). Subsequently, after 14 d, dorsal skin sections were subjected to hematoxylin and eosin (H&E) staining. EF-2001 treatment significantly promoted hair-follicle conversion from telogen to anagen compared with the control, and this conversion was comparable to the levels observed with minoxidil treatment (Figure 3A). The graph in Figure 3B shows the thickness of the skin in each treatment group. The skin thicknesses of the EFH- and minoxidil-treated groups were similar. The EFL- and EFH-treated groups showed a clear increase in skin thickness compared with that of the EFC-treated group. The skin thicknesses were 246.65 ± 8.34, 610.81 ± 185.11, 1010.30 ± 77.72, 1101.16 ± 48.28, and 1171.02 ± 162.74 μm in the control and EFC-, EFL-, EFH-, and minoxidil-treated groups, respectively.

### 2.3. Effect of EF-2001 on the Expression of FGF2, FGF7, and FGF10 mRNA in C57BL/6 Male Mice

As shown in Figure 4, the expression of *FGF7* in the minoxidil-treated group was twice of that in the EFC-treated group. However, *F**GF2* and *F**GF10* expression in the minoxidil-treated group did not differ from that in the EFC-treated group. The expression of *F**GF2* and *F**GF10* in the EFL- and EFH-treated groups was higher than that in the EFC-treated group, while *F**GF7* expression was similar in the EFL-, EFH-, and EFC-treated groups. The mRNA expression of *FGF**2* was 1.02 ± 0.24, 1.95 ± 0.25, 1.57 ± 0.37, and 1.13 ± 0.20 for *FGF**2*, 1.07 ± 0.41, 1.34 ± 0.23, 1.26 ± 0.49, and 2.14 ± 0.78 for *FGF**7*, and 1.04 ± 0.35, 3.43 ± 0.76, 4.27 ± 0.83, and 1.06 ± 0.42 for *FGF**10* in the EFC-, EFL-, EFH-, and minoxidil-treated groups, respectively.

### 2.4. Effect of EF-2001 on the Expression of Wnt5a, Wnt5b, and Wnt10b mRNA in C57BL/6 Male Mice

As shown in Figure 5, the expression of *Wnt10b* in the EFL- and EFH-treated groups was higher than that in the EFC-treated group, while *Wnt5b* expression was similar in the EFL-, EFH-, and EFC-treated groups. In the minoxidil-treated group, the expression of *Wnt5a* and *Wnt10b* increased compared with that in the EFC-treated group. The mRNA expression was 1.06 ± 0.37, 1.45 ± 0.21, 1.54 ± 0.53, and 1.82 ± 0.34 for *Wnt**5a*, 1.01 ± 0.16, 1.16 ± 0.12, 1.01 ± 0.30, and 1.79 ± 2.24 for *Wnt**5b**,* and 1.05 ± 0.34, 2.45 ± 0.36, 2.20 ± 0.53, and 4.93 ± 0.86 for *Wnt10b* in the EFC-, EFL-, EFH-, and minoxidil-treated groups, respectively.

### 2.5. Effect of EF-2001 on the Expression of VEGF-A and VEGF-B mRNA in C57BL/6 Male Mice

As shown in Figure 6, among the groups treated with EF-2001, the expression of *V**EGF-A* was significantly higher in the EFH-treated group, and that of *V**EGF-B* was significantly higher in the EFL-treated group compared with that in the EFC-treated group. In the minoxidil-treated group, the expression of the *V**EGF-B* increased compared with that in the EFC-treated group. The mRNA expression was 1.02 ± 0.12, 1.05 ± 0.28, 1.66 ± 0.39, and 1.16 ± 0.20 for *V**EGF-A* and 1.05 ± 0.38, 6.01 ± 1.26, 16.99 ± 16.14, and 38.48 ± 26.93 for *V**EGF-B* in the EFC-, EFL-, EFH-, and minoxidil-treated groups, respectively.

### 2.6. Effect of EF-2001 on the Expression of IGF1 and IGF1-R mRNA in C57BL/6 Male Mice

As shown in Figure 7, the expression of *IGF1-R* was similar in the EFC-, EFL-, and EFH-treated groups. *IGF1* expression significantly decreased in the EFL- and EFH-treated groups. In the minoxidil-treated group, the expression of *IGF1* and *I**GF1-R* increased compared with that in the EFC-treated group. The mRNA expressions were 1.01 ± 0.22, 0.68 ± 0.13, 0.56 ± 0.11, and 1.45 ± 0.27 for *IGF1* and 1.08 ± 0.47, 0.93 ± 0.11, 1.31 ± 0.37, and 2.00 ± 0.51 for *IGF1-R* in the EFC-, EFL-, EFH-, and minoxidil-treated groups, respectively.

## 3. Discussion

DP cells regulate hair regrowth and secrete multiple signaling factors that control the hair cycle [35]. Generally, hair follicles in people with androgenetic alopecia symptoms have intact hair-follicle stem cells, and DP signaling transmission is insufficient to stimulate the initiation of the hair cycle [36]. Therefore, restoring DP cells, which can induce new hair, is a potential treatment for hair loss. However, to date, the effects of EF-2001 on hair growth have not been reported in vivo or in vitro.

The proliferation of DP cells is essential for hair-follicle morphogenesis and growth. In this study, we established that EF-2001 induced the proliferation of DP cells at a rate similar to that seen with minoxidil treatment (Figure 1). Moreover, EF-2001 treatment in C57BL/6 male mice for 14 d led to macroscopic hair growth. Hair growth in the shaved region was observed in minoxidil-, EFL- (5 mg/mL), and EFH-treated (50 mg/mL) groups on day 6 and increased by the 14th day (Figure 2).

To confirm the effect of EF-2001 on hair follicles, skin tissues were stained using H&E, and hair-follicle depth and dermis thickness were observed (Figure 3). During the anagen phase, melanin synthesis begins, thickening the dermis layer, increasing the size of hair follicles, and making the dermis and subcutaneous fat deeply characterized. In anagen stage II, the hair follicles are located only in the dermal layer and do not reach the subcutaneous fat layer, but in anagen stage III, the hair follicles reach the subcutaneous fat layer [37]; the increased hair-follicle length decreases again during the catagen stage. Therefore, measuring changes in the length and thickness of the dermis and subcutaneous tissue due to the continuous growth of anagen hair follicles and degeneration of catagen hair follicles is of great significance in determining the hair cycle. Minoxidil treatment led to a significant increase in the thickness of the skin compared with the control treatment, and thickness also increased remarkably after EFL (5 mg/mL) and EFH (50 mg/mL) treatment. Therefore, EF-2001 improves hair growth by extending the anagen phase in the hair cycle.

DP cells release growth factors and induce hair growth and regrowth [38,39]. Therefore, the expression of marker genes and growth factors was analyzed using quantitative-polymerase chain reaction (qPCR) in mice skin tissues. *FGF7* and *FGF10* play a role in maintaining the anagen phase over long periods. Furthermore, *FGF2* modulates *FGF7* and *FGF10* and plays a role in increasing DP cell proliferation to maintain DP cell proliferation and the anagen phase for extended periods (Figure 4).

Wnt signaling in DP cells plays an essential role in regulating hair-follicle morphogenesis, differentiation, and follicle regeneration. EF-2001 is considered to affect *Wnt10b* expression and may contribute to new hair growth (Figure 5).

VEGF promotes angiogenesis around hair follicles and induces hair anagen [40]. EF-2001 treatment modulated *VEGF-A* and *VEGF-B* expression, promoting angiogenesis and inducing the anagen phase of hair (Figure 6).

IGF1 is an important growth factor that regulates hair growth. IGF1 is produced only in mesenchymal cells, and its receptors are present in hair-follicle cells. Therefore, secreted IGF1 promotes the proliferation of hair-follicle cells and significantly increases the length of the hair-follicle tissue [10]. However, *IGF1* decreased after treatment with EF-2001; therefore, this phenomenon needs to be studied further (Figure 7).

In vitro experiments confirmed that EF-2001 induced the proliferation of DP cells. Furthermore, in in vivo experiments, EF-2001 accelerated hair regrowth and promoted hair-follicle transformation from telogen to anagen, affecting the expression levels of several growth-factor genes (*F**GF* and *V**EGF*) and hair-growth-marker genes (*Wnt*) in mice (Figure 8). In conclusion, EF-2001 showed a hair-growth effect comparable to that of minoxidil.

Moreover, EF-2001 use did not show any side effects in mice during the course of our experiment. To use EF-2001 as a treatment for hair loss for a prolonged period, it is necessary to confirm that its side effects with prolonged usage are fewer than those of minoxidil. It is also necessary to confirm the correlation between growth factors and marker genes to determine the underlying mechanism of action of EF-2001 and to compare the protein expression levels of the investigated genes.

## 4. Materials and Methods

### 4.1. Materials

The MTT assay kit was purchased from Duchefa Biochemie (Harlem, The Netherlands). Penicillin, streptomycin, and fetal bovine serum (FBS) were obtained from Capricorn (Ebsdorfergrund, Germany). Dimethyl sulfoxide (DMSO), minoxidil, TRI-reagent, olive oil, and trypsin-EDTA solution were purchased from Sigma-Aldrich (St. Louis, MO, USA). Acetone (purity 99.8%) was purchased from Daejung Chemicals & Metals Co. (Siheung, Korea). Phosphate-buffered saline (PBS) was purchased from Gibco Life Technologies (Rockville, MD, USA). A 5% minoxidil solution was obtained from Dongsung Pharm. Co. (Seoul, Korea).

### 4.2. EF-2001 (Enterococcus faecalis-2001)

EF-2001 originated from human feces is a merchantable-quality parabiotic purified by Bereum Co., Ltd. (Wonju, Korea) and is supplied as a heat-killed, dried powder. Before being heat-killed, dried EF-2001 contained 7.5 × 10^12^ units per gram [34].

### 4.3. Cell Culture

Human dermal papilla cells (HDPCs) were obtained from CEFO Co., Ltd. (cat. no. CB-HDP-001; Seoul, Korea). HDPCs were cultured in Dulbecco’s modified Eagle’s medium (DMEM; Sigma-Aldrich) supplemented with 10% (*v*/*v*) FBS and penicillin–streptomycin (100 µg/mL). HDPCs were incubated at 37 °C in a humidified atmosphere containing 5% CO_2_. Trypsin-EDTA solution was used to detach cells. HDPCs between passages 4 and 8 were used for all experiments.

### 4.4. Cell-Proliferation Assay

HDPC proliferation was determined using the MTT assay. HDPCs (5.0 × 10^4^ cells/mL) were seeded into 96-well cell-culture plates in DMEM supplemented with 10% FBS at 37 °C for 24 h. After separating the supernatant, the HDPCs were washed twice with PBS. The medium was then replaced with serum-free DMEM containing EF-2001 (0, 50, 75, 150, 250, and 500 µg/mL) and 3 µM minoxidil, and the HDPCs were incubated for 48 h. Thereafter, 20 µL of MTT (5 mg/mL) was added to each well, and HDPCs were incubated for 4 h at 37 °C. The supernatant was then removed, and 200 µL of DMSO was added to dissolve the MTT formazan products. The absorbance was measured at 595 nm using an ELx800 microplate reader (BioTek Instruments Inc., Winooski, VT, USA).

### 4.5. Animal Experiments

Male C57BL/6 mice (6-week-old) were purchased from Central Lab. Animal, Inc. (Seoul, Korea). The mice underwent a 14 d acclimation period for maintenance at 20–22 °C and 40–50% relative humidity with a controlled light–dark cycle. Food and water were provided ad libitum. The Institutional Animal Care and Use Committee (IACUC, YWCI-202003-005-01) of Yonsei University (Wonju, Korea) approved the protocol for this study. All steps to minimize pain in the animals were taken. The animals were divided into five randomized groups (*n* = 7) to study hair-growth-promoting activity. After acclimation, the dorsal hair of the mice was shaved using an electric shaver under isoflurane (Hana Pharm Co., Hwaseong, Korea) anesthesia. Next, the depilatory Niclean (Ildong Pharmaceutical Co., Ltd., Seoul, Korea) was applied to the backs of mice. After 48 h, hair was shaved, and the depilated area was treated with 200 µL each of the following: control: PBS, minoxidil: 5% minoxidil solution, EFC: 0 mg/mL EF-2001 in 3:1 mixture of acetone and olive oil, EFL: 5 mg/mL EF-2001 in 3:1 mixture of acetone and olive oil, and EFH: 50 mg/mL in 3:1 mixture of acetone and olive oil) once a day for 14 d (Figure 9).

### 4.6. H&E Staining

Hematoxylin and eosin (H&E) staining was performed as previously described [41]. Briefly, dorsal skin lesions were obtained from the control (PBS), minoxidil (5%), and EF-2001 (0, 5, and 50 mg/mL) groups, fixed in 10% formalin, and stabilized in 10, 15, and 20% sucrose solution. To embed the fixed skin in a cryomold, the FSC 22 Clear frozen section compound (Cat. no. 3801480; Leica Biosystems, Wetzlar, Germany) was used. Afterward, the cryo-tissues were sectioned, stained using H&E, visualized under an Olympus DP80 microscope, and analyzed using imaging software (Cell Sens 1.8; Olympus Europa SE & Co., KG, Hamburg, Germany).

### 4.7. RNA Preparation and qPCR

RNA preparation, qPCR, and primer sequencing were performed as previously described [41]. Total RNA was isolated from tissues extracted from the dorsal lesions of mice using TRI-reagent. The total RNA concentration was determined using a Colibri Microvolume spectrophotometer (Titertek Berthold, Pforzheim, Germany). Further, total RNA was used as a template for cDNA synthesis using a cDNA synthesis kit (Takara Bio, Shiga, Japan). qPCR was performed using SYBR Green I and a LightCycler 96 instrument (Roche, Basel, Switzerland). The cycling conditions were as follows: one cycle of denaturation at 95 °C for 10 min, followed by 45 cycles of denaturation at 95 °C for 10  s, annealing at 55 °C for 10  s, and extension at 72 °C for 10  s. The Cq value for each reaction was determined using LightCycler 96 SW1.1 software. The expression levels of the analyzed genes were normalized to that of GAPDH. The primers used are presented in Table 1.

### 4.8. Statistical Analysis

Experimental results are expressed as mean ± standard deviation (SD). One-way analysis of variance (ANOVA) was used for multiple comparisons, followed by Tukey–Kramer’s post hoc analysis. *p* < 0.05, *p* < 0.01, and *p* < 0.001 were considered statistically significant, highly significant, and very highly significant, respectively.

## Figures and Tables

**Figure 1 ijms-23-05413-f001:**
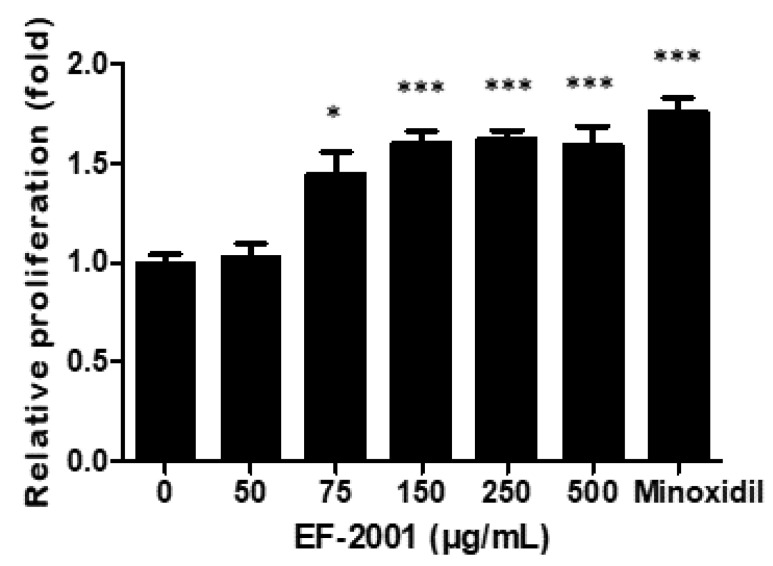
Effect of minoxidil and EF-2001 on the proliferation of human dermal papilla cells (HDPCs) determined by MTT assay. Minoxidil (3 μM) and EF-2001 (at various concentrations) induced the proliferation of HDPCs, measured by determining optical density at 595 nm using a microplate reader. Data are expressed as the mean ± SD (*n* = 4). ***
*p* < 0.05, *****
*p* < 0.001 versus control.

**Figure 2 ijms-23-05413-f002:**
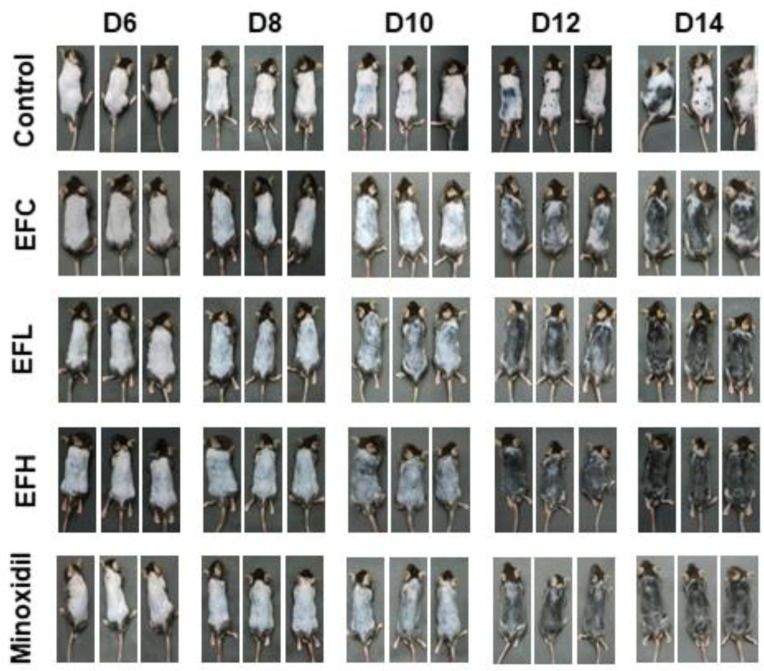
Comparison of hair regrowth in C57BL/6 male mice after topical application of phosphate-buffered saline (PBS), minoxidil (5%), or EF-2001 (0, 5, 50 mg/mL) for 6, 8, 10, 12, and 14 d. Hair growth on the dorsal skin of C57BL/6 mice after treatment for 14 d with control (PBS, *n* = 7), EFC (solvent control, 0 mg/mL of EF-2001 in 3:1 mixture of acetone and olive oil), EFL (EF-2001 low concentration, 5 mg/mL of EF-2001 in 3:1 mixture of acetone and olive oil), EFH (EF-2001 high concentration, 50 mg/mL of EF-2001 in 3:1 mixture of acetone and olive oil), and minoxidil (5% minoxidil, *n* = 7). Dorsal skin was photographed at 6, 8, 10, 12, and 14 d of treatment. Images were obtained using a Galaxy S10 Plus camera (Samsung, Seoul, Korea).

**Figure 3 ijms-23-05413-f003:**
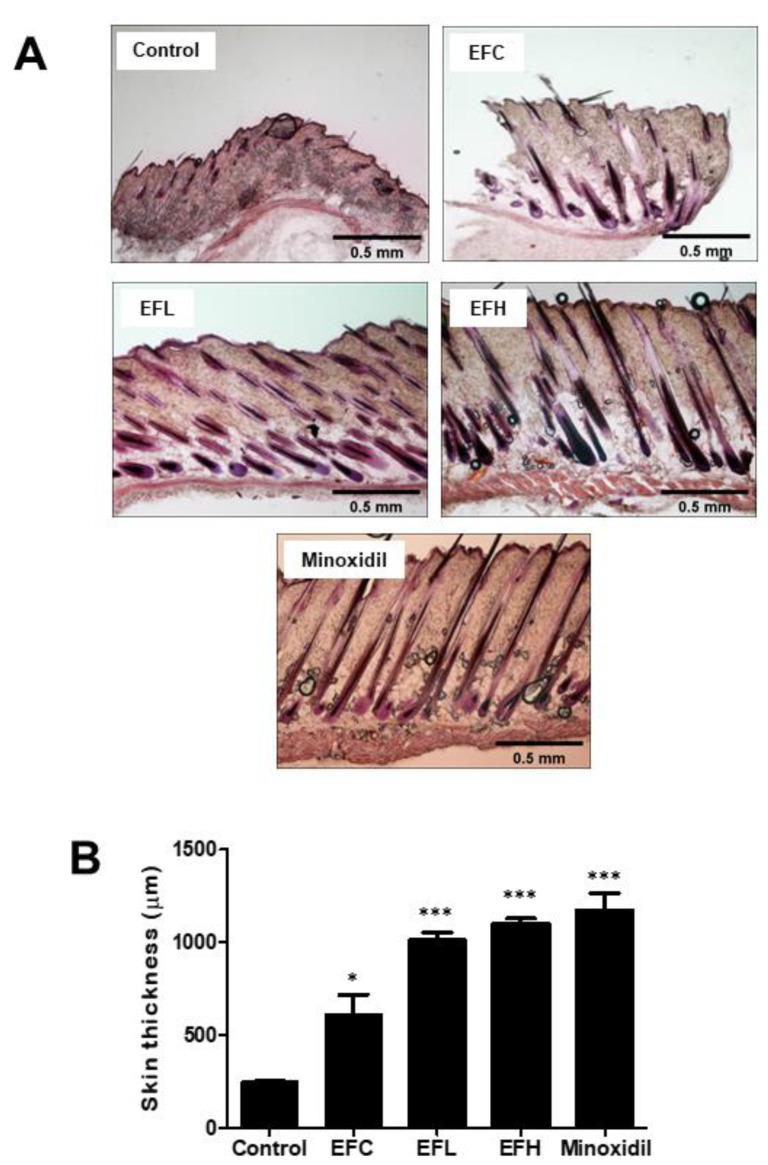
The effect of hair regrowth treatments on C57BL/6 male mice skin as assessed by hematoxylin and eosin (H&E) staining. (**A**) Longitudinal sections of the dorsal skins for each group by H&E staining. (**B**) The skin thickness in the section for each group. Data are expressed as mean ± SD (*n* = 3). ***
*p* < 0.05, *****
*p* < 0.001 versus control.

**Figure 4 ijms-23-05413-f004:**
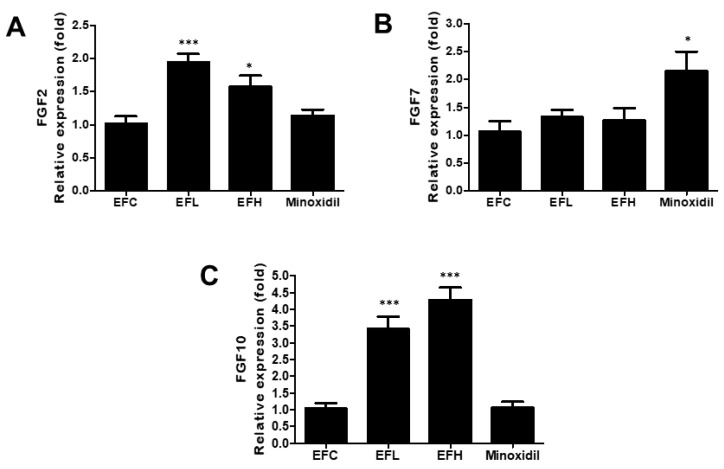
The effect of EF-2001 on the mRNA expression of hair-growth-related genes, (**A**) *FGF2*, (**B**) *FGF7*, and (**C**) *FGF10*, in C57BL/6, male mice. Quantitative PCR analysis of the mRNA expression of *FGF2*, *FGF7*, and *FGF10* in the EFC-, EFL-, EFH-, and minoxidil-treated groups. Data are expressed as mean ± SD (*n* = 5). ***
*p* < 0.05, *****
*p* < 0.001 versus EFC.

**Figure 5 ijms-23-05413-f005:**
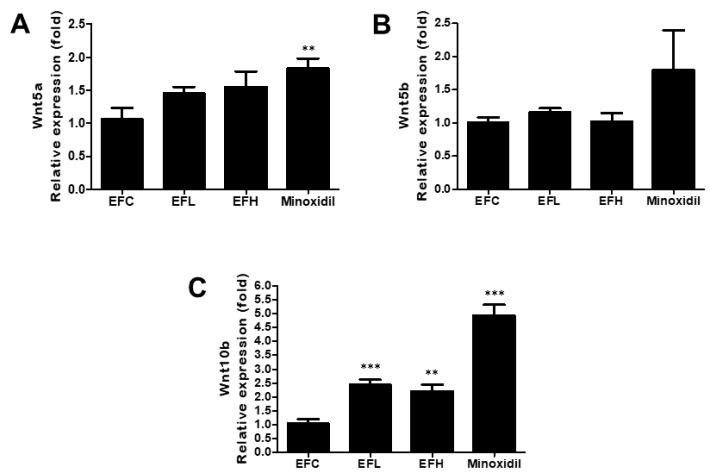
The effect of EF-2001 on the mRNA expression of hair-growth-related genes, including (**A**) *Wnt5a,* (**B**) *Wnt5b,* and (**C**) *Wnt10b*, in C57BL/6 male mice. Quantitative PCR analysis of the mRNA expression of *Wnt5a, Wnt5b,* and *Wnt10b* in the EFC-, EFL-, EFH-, and minoxidil-treated groups. ***** p* < 0.01, *****
*p* < 0.001 versus EFC. Data are expressed as mean ± SD (*n* = 5).

**Figure 6 ijms-23-05413-f006:**
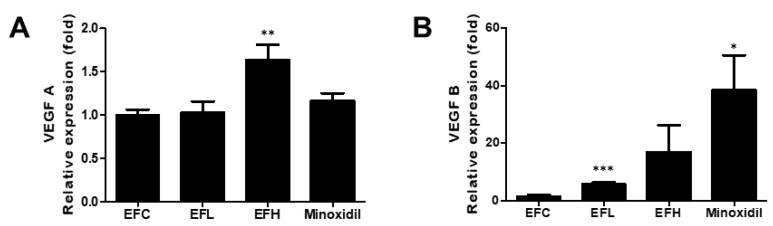
The effect of EF-2001 on the mRNA expression of hair-growth-related genes (**A**) *V**EGF-A* and (**B**) *V**EGF-B* in C57BL/6 male mice. Quantitative PCR analysis of the mRNA expression of *V**EGF-A* and *V**EGF-B* in the EFC-, EFL-, EFH-, and minoxidil-treated groups. ***
*p* < 0.05, ***** p* < 0.01, *****
*p* < 0.001 versus EFC. Data are expressed as mean ± SD (*n* = 5).

**Figure 7 ijms-23-05413-f007:**
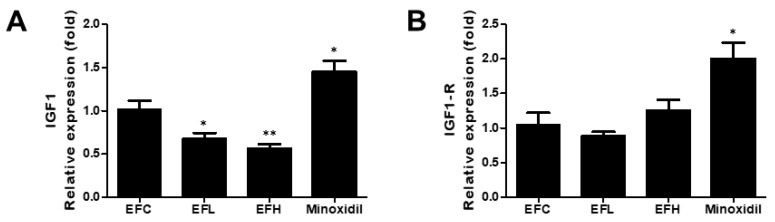
The effect of EF-2001 on the mRNA expression of hair-growth-related genes (**A**) *I**GF1* and (**B**) *I**GF1-R* in C57BL/6 male mice. Quantitative PCR analysis of the mRNA expression of *I**GF1* and *I**GF1-R* in the EFC-, EFL-, EFH-, and minoxidil-treated groups. ***
*p* < 0.05, ***** p* < 0.01 versus EFC. Data are expressed as mean ± SD (*n* = 5).

**Figure 8 ijms-23-05413-f008:**
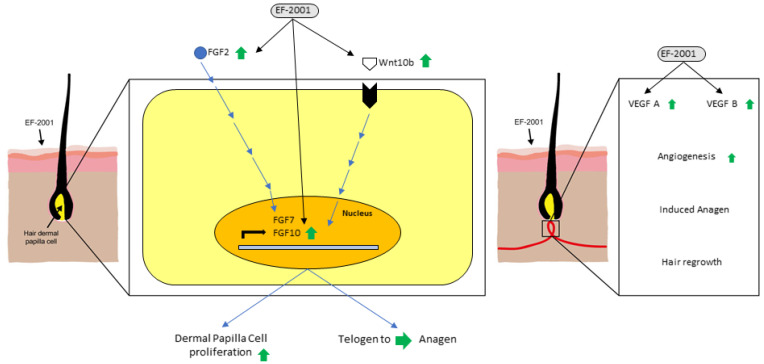
EF-2001 affects hair regrowth in vitro and in vivo.

**Figure 9 ijms-23-05413-f009:**
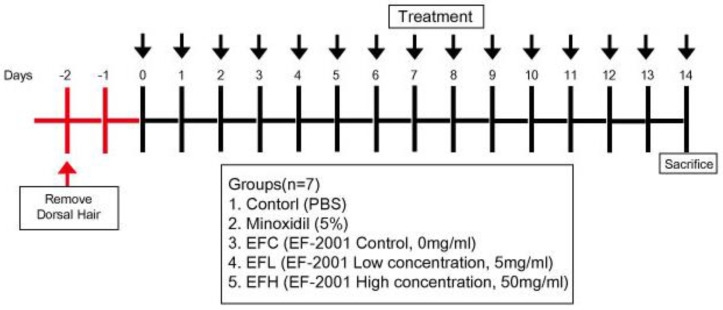
Schematic diagram of experimental protocol in C57BL/6 mice.

**Table 1 ijms-23-05413-t001:** Primer sequences.

Species	Primer	Forward	Reverse
Mouse	*Igf1*	5′-GTCGTCTTCACACCTCTTCTACCT-3′	5′-GCACAGTACATCTCCAGTCTCCT-3′
	*Igf1r*	5′-CTCAGGCTTCATCCGCAACAG-3′	5′-GTTCTCCAACTCCGAGGCAATG-3′
	*Wnt5a*	5′-CTGGCAGGACTTTCTCAAGG-3′	5′-CTCTAGCGTCCACGAACTCC-3′
	*Wnt5b*	5′-TCGGAGGAGCAGGGCCGAGC-3′	5′-CAGCTTGCCCTGGCGGGTGA-3′
	*Wnt10b*	5′-CCTGTCCGGACTGAGTAAGC-3′	5′-TTGCTCACCACTACCCTTCC-3′
	*Fgf2*	5′-CAAGAACGGCGGCTTCTTC-3′	5′-GAAAGAAACAGTATGGCCT-3′
	*Fgf7*	5′-AGACTGTTCTGTCGCACC-3′	5′-CCGCTGTGTGTCCATTTAG-3′
	*Fgf10*	5′-TGTCCGCTGGAGAAGGCTGTTC-3′	5′-CTATGTTTGGATCGTCATGG-3′
	*Vegfa*	5′-CGAGATAGAGTACATCTTCAAGCC-3′	5′-TCATCGTTACAGCAGCCTGC-3′
	*Vegfb*	5′-AAAAAAAAAGGAGAGTGCTGTGAAG-3′	5′-TCCCAGCCCGGAACAGA-3′
	*Gapdh*	5′-GCCAAGGTCATCCATGACAACT-3′	5′-GAGGGGCCATCCACAGTCTT-3′

## Data Availability

The data used to support the findings of this study are available from the corresponding author upon request.

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
