# Peer review of "Heat-Killed Enterococcus faecalis EF-2001 Induces Human Dermal Papilla Cell Proliferation and Hair Regrowth in C57BL/6 Mice"

_ijms, 2022, doi:10.3390/ijms23105413_

Round 1

Reviewer 1 Report

In the manuscript entitled "Heat-killed Enterococcus faecalis EF-2001 Induces the Proliferation of Human Dermal Papilla Cells and Hair Regrowth in C57BL/6 Mice", the Authors reported the beneficial effects of heat-killed Enterococcus faecalis EF-2001 on hair loss prevention and regrowth using human dermal papilla cells and male C57BL/6 mice. The topic is interesting but some issues should be addressed:

- In Figure 3, the image of the control group seems at different magnification; the Authors should add a ruler in the frame of each image (each experimental group) to demonstrate the same magnification.

Author Response

Response to Reviewer Comments 1.

Thank you very much for your comments. We have corrected the manuscript according to your comments, and our responses and changes made in the manuscript are as follows.

In Figure 3, the image of the control group seems at different magnification; the Authors should add a ruler in the frame of each image (each experimental group) to demonstrate the same magnification.

Response: Thanks for the reviewer's comments. The images of the control group are at the same magnification. Therefore, according to the reviewer's opinion, it was replaced with a picture with a ruler in the frame.

Reviewer 2 Report

The study was well planned and carried out in a cohesive way. The work still needs some changes and clarifications after the changes this work can be accepted for publication.

  1. Line 69-70, the sentence seems to be incomplete.
  2. Line 72- ‘reported’ word spell error.
  3. Line 74-‘been’ word spell error.
  4. In this study the concentration of EF-2001 was set from 50 to 500 µg/mL. Any previous works standardized the minimum concentration of EF-2001 as 50 µg/mL or lesser than this.
  5. In results 2.1, the higher concentration of 500 µg/mL showed significantly less effect than 250µg/mL. Justify the result.
  6. In material methods 4.5; line 309, the author can explain how 5% minoxidil is prepared.
  7. Line 310; need explanation for the group EFC (solvent control 0 mg/mL). If it is taken as the EFC – EF-2001 is not in the solvent, then it is like the control. Either author can explain what makes EFC (solvent control 0 mg/mL) different from the control PBS. The same need to be explained in the results 2.2, line 103-105.How EFC (solvent control 0 mg/mL) showed more hair growth than the untreated control. And the EFC has been compared in the rest of the results also. Hence proper justification is needed for EFC and the comparison between EFC-EFL-EFH.
  8. In line 310-311, the author explained about the EF concentration is as low as 5mg/mL and high as 50mg/mL. Can the author explain the concentration, how this was selected for this study, any previous reports supported the concentration? Justify the concentrations of EFL and EFH.
  9. In line 100 & 105, it was given as Figure 2A and Figure 2B. But neither in the Figure nor in its legend, it was not marked as 2A and 2B.
  10. In figure, 3C need not be given separately. The comparison among EC-EFL-EFH is meaningless. The comparison in 3A is enough. The author can decide whether to keep 3C in the manuscript. Same to be checked in results 2.2 lines 109-112.
  11. Author can explain about fgf2, fgf7, fgf10, wnt5a, wnt5n and wnt10b, vegfa, vegfb, igf1 and igf1r in the introduction in a few lines. The genes directly introduced in the results 2.3 and 2.4 (but, later explained in discussion) the lines 135-137, 145-148, 154, 161-164 can be placed in introduction section.
  12. In figure 4, A-B, C-D, E-F graphs can be combined. All the five groups Control-Minoxidil-EFC-EFL-EFH graph can be given in a single graph. So, three graphs can be given for fgf2, fgf7, fgf10. The same goes for figures 5, 6and 7.
  13. Author can confirm the expression of any one of the sets of proteins (FGF2, FGF7, FGF10 or WNT5A, WNT5B and WNT10B or VEGFA, VEGFB or IGF1 and IGF1R) in western blotting to substantiate the -PCR results.

Author Response

Response to Reviewer Comments 2

Thank you very much for your comments. We have corrected the manuscript according to your comments, and our responses and changes made in the manuscript are as follows.

  1. Line 69-70, the sentence seems to be incomplete.

Response: We checked the sentence. and corrected.

  1. Line 72- ‘reported’ word spell error.

Response: We have confirmed the error. The word was corrected.

  1. Line 74-‘been’ word spell error.

Response: We have confirmed the error. The word was corrected.

  1. In this study the concentration of EF-2001 was set from 50 to 500 µg/mL. Any previous works standardized the minimum concentration of EF-2001 as 50 µg/mL or lesser than this.

Response: We check the cytotoxicity of our samples when we do in vitro cell experiments. This is because the cytotoxicity of the same sample is different for each cell. We performed the cytotoxicity of EF-2001. Efficacy was confirmed up to the maximum concentration of 500 µg/mL without toxicity. So it was set from 50 to 500 µg/mL.

  1. In results 2.1, the higher concentration of 500 µg/mL showed significantly less effect than 250µg/mL. Justify the result.

Response: We have confirmed the opinions of reviewers. After receiving the reviewer's opinion, significance was tested again. We found significance at 500 μg/mL as ***P < 0.001. So we corrected our mistake. And the resulting figures are described in the text.

  1. In material methods 4.5; line 309, the author can explain how 5% minoxidil is prepared.

Response: 5% minoxidil, which we set as a control group, was used as a marketed therapeutic product. The concentration contained 5% minoxidil, so 5% was used as a positive control.

  1. Line 310; need explanation for the group EFC (solvent control 0 mg/mL). If it is taken as the EFC – EF-2001 is not in the solvent, then it is like the control. Either author can explain what makes EFC (solvent control 0 mg/mL) different from the control PBS. The same need to be explained in the results 2.2, line 103-105.How EFC (solvent control 0 mg/mL) showed more hair growth than the untreated control. And the EFC has been compared in the rest of the results also. Hence proper justification is needed for EFC and the comparison between EFC-EFL-EFH.

Response: EFC treated only the solvent (3:1 mixture of acetone and olive oil) used to deliver EF-2001 subcutaneously. Therefore, EFC was compared with EFL and EFH. In addition, PBS was carried out by creating an additional group as a control to confirm the effect of the solvent (3:1 mixture of acetone and olive oil). It has been added to the text.

  1. In line 310-311, the author explained about the EF concentration is as low as 5mg/mL and high as 50mg/mL. Can the author explain the concentration, how this was selected for this study, any previous reports supported the concentration? Justify the concentrations of EFL and EFH.

Response: So far, the experimental results of EF-2001 were the results of a study on oral administration. Therefore, there are no prior reports of experiments with EF-2001 applied to the skin. This is our first experiment. Therefore, the lowest concentration was 5 mg/mL, and the 10-fold higher concentration was 50 mg/mL. The concentration was arbitrarily set to a high concentration.

  1. In line 100 & 105, it was given as Figure 2A and Figure 2B. But neither in the Figure nor in its legend, it was not marked as 2A and 2B.

Response: I have confirmed your opinion. It is our mistake. So it was corrected.

  1. In figure, 3C need not be given separately. The comparison among EC-EFL-EFH is meaningless. The comparison in 3A is enough. The author can decide whether to keep 3C in the manuscript. Same to be checked in results 2.2 lines 109-112.

Response: We have confirmed your opinion. And 3C was deleted because we agreed to the deletion of Figure 3C.

  1. Author can explain about fgf2, fgf7, fgf10, wnt5a, wnt5n and wnt10b, vegfa, vegfb, igf1 and igf1r in the introduction in a few lines. The genes directly introduced in the results 2.3 and 2.4 (but, later explained in discussion) the lines 135-137, 145-148, 154, 161-164 can be placed in introduction section.

Response: Thank you for the reviewer's comments. According to the opinions, the contents were moved to Introduction.

  1. In figure 4, A-B, C-D, E-F graphs can be combined. All the five groups Control-Minoxidil-EFC-EFL-EFH graph can be given in a single graph. So, three graphs can be given for fgf2, fgf7, fgf10. The same goes for figures 5, 6and 7.

Response: Thank you for the reviewer's comments. I combined the graphs according to the comments and made a graph again.

  1. Author can confirm the expression of any one of the sets of proteins (FGF2, FGF7, FGF10 or WNT5A, WNT5B and WNT10B or VEGFA, VEGFB or IGF1 and IGF1R) in western blotting to substantiate the -PCR results.

Response: We agree with you. However, this study focused on gene expression. Future thesis preparation will try to elucidate signal transduction by protein expression. I hope for your understanding.

Author Response

Response to Reviewer Comments 3

Thank you very much for your comments. We have corrected the manuscript according to your comments, and our responses and changes made in the manuscript are as follows.

  1. High similarity index, mainly in the Materials and Methods section

According to “iThenticate”, the similarity index is high, mainly in the materials and methods section.

Response: We reviewed the similarity and deleted and corrected a lot of the content.

  1. Introduction

Line 34: add the link for the “Health Insurance Review and Assessment Service”, related to the presented data.

Response: We used [4,5] reference. So the sentence was corrected.

Line 42: give examples for the symptoms.

Response: We added an example and modified it to a specific sentence.

Line 68: abbreviate the scientific name (E. faecalis).

Response: We have changed it to Scientific name.

  1. Results/ Discussion (including Figures)
  • In general, the legend of most figures repeat data, already presented in the Materials and Methods or results section.

Response: We modified the figure legend to avoid repetition with Materials and Methods.

  • Figure 1: it is better to present the results (Y-axis) as proliferation fold (as mentioned in 2.1)

Response: We recreated Figure 1 graph by changing to fold as per your opinion.

  • Line 100 and 105: in Figure 2, there is no A and B.

Response: We have confirmed your opinion. It is our mistake. So we corrected that part.

  • Line 127-128: the measurement method should be in the Materials and Methods section.

Response: I agree with your opinion. So we deleted that part.

  • Figure 4: combine the results of each gene in one graph (A & B; C & D; E & F).

Response: Thank you for the reviewer's comments. I combined the graphs according to the comments and made a graph again. And the numerical value of the exact result is expressed in the text.

  • Figure 5: combine the results of each gene in one graph (A & B; C & D; E & F).

Response: Thank you for the reviewer's comments. I combined the graphs according to the comments and made a graph again. And the numerical value of the exact result is expressed in the text.

  • Figure 6: combine the results of each gene in one graph (A & B; C & D).

Response: Thank you for the reviewer's comments. I combined the graphs according to the comments and made a graph again. And the numerical value of the exact result is expressed in the text.

  • Figure 7: combine the results of each gene in one graph (A & B; C & D).

Response: Thank you for the reviewer's comments. I combined the graphs according to the comments and made a graph again. And the numerical value of the exact result is expressed in the text.

  1. Materials and Methods
  • Line 287, do you mean 5.0 x 104 ?

Response: I have confirmed your comment. We fixed that error.

  • Figure 9 was not mentioned in the text.

Response: I have confirmed your comment. We added Figure 9.

  • Add a reference for section 4.6.

Response: We added a reference.

  • Line 328: add a link or a reference for the manufacturer's instructions.

Response: we added a reference

  • Add references for the used primers and present the details in a table

Response: We have changed the Primer information into a table format. And we added a reference to the primer design

.• For statistical analysis, P-values < 0.05 were considered statistically significant.

Response: I have confirmed your comment. We corrected that sentence.

  1. Informed Consent Statement

Informed consent was obtained from which subjects?

Response: I have confirmed your comment. We added content.

  1. References

Ref.4: Correct the journal name style.

Response: I have modified it according to your opinion.

Ref. 24, 28-33 and 34: Italicize the scientific name

Response: I have modified it according to your opinion.

Round 2

Reviewer 1 Report

The ruler in the images of Figure 3 is not clear. An example of what it should be done is present here: https://images.app.goo.gl/3BBWU3qK4JZhaG7e8

Author Response

I have confirmed your opinion. We understood what you were proposing. We have enlarged the photo for the reader. And a scale bar was clearly added. And we checked the English spells carefully. 

Reviewer 2 Report

 Accept in present form

Author Response

Thank you for your decision.

Round 3

Reviewer 1 Report

How do the Authors explain the increased thickness of the smooth muscle layer beneath the derma in EFH and Minoxidil group?

Author Response

Response: Thanks for your nice question. We thought about your question.

First of all, the skin is composed of the epidermis, dermis, and subcutaneous tissue. The epidermis is composed of keratinocytes, melanocytes, Langerhans cells, and merkel cells. In particular, the dermis is rich in collagen, elastin, and hyaluronic acid, and plays an important role in matrix composition. It is also a part that receives nutrients because of the distribution of peripheral blood vessels.

Many studies have reported that minoxidil induces stimulation of various growth factors, which in turn affects collagen and elastin. Also, the scalp becomes thinner in the symptoms of hair loss. There is a report that the thickness of the scalp becomes thicker when hair loss is restored. Minoxidil has the effect of helping hair growth by increasing the supply of nutrients to the hair roots by expanding the blood vessels in the scalp and expanding capillaries.

The exact mechanism for scalp health is not yet known, but we will explain the results of previous studies and our thoughts. According to the current experimental results, the phenomenon of skin thickening by minoxidil is that minoxidil induces various growth factors, and stimulates various nutrition supply through vasodilation by minoxidil, and the effect of collagen and elastin in the papillary layer of the dermis. I think that the thickness of the skin improves through the recovery of the dermal layer by the grown hair follicles.

In addition, it is hypothesized that EF-2001 exhibits some different growth factor gene expression patterns than minoxidil, but it induces effects on collagen accumulation or elastin to show the same skin thickness recovery effect.

Even from our tissue staining results, it can be confirmed that the dermal layer is thick and evenly arranged. The hair shaft looks clean in the shape of a ladder in the middle of the hair follicles, and it can be seen that the EFH and minoxidil applied groups have longer and larger hair follicles compared to the control group. Hair follicles grown in the papillary layer of the dermis go through the catagen phase and become close to the epidermis, and it is thought that the dermal papilla cells in the hair follicle form a new hair growth cycle. Hair growth regulation of dermal papilla cells was found to be related to the size of the dermal papilla. Regarding the hair follicle cycle, it is known that dermal papilla cells, which were atrophied in the degenerative phase, enlarge and form large hair bulbs through proliferation of epithelial cells to cause hair growth.

However, it is true that it is difficult to explain with the results of comparative studies of current gene expression patterns. We are thinking and pushing forward as the next research project to elucidate the mechanism. Your opinion is very important, and the molecular mechanism of EF-2001 as well as minoxidil will be clarified in the next research project.

[Reference]

Botchkarev BO, Kishimoto J, Molecular control of epithelial mesenchymal interactions during hair follicle cycling, J. Invest. Dermatol. Symp. Proc, 8:46-53 (2003).

Cotsarelis G, Millar SE, Towards a molecular understanding of hair loss and its treatment, Trends Mol. Med, 7:293-301 (2001).

Seven MR, Bori H, Carina VDV, Stefan E, Kerstin F, Ian AM, Kurt SS, Ralf P, A comprehensive guide for the accurate classification of murine hair follicles in distinct hair cycle stages, J. Invest. Dermatol, 117: 3-15 (2001).

Fhayli W, Boëté Q, Harki O, Briançon-Marjollet A, Jacob MP, Faury G. Rise and fall of elastic fibers from development to aging. Consequences on arterial structure-function and therapeutical perspectives

Matrix Biol. 84:41-56 (2019).

Slove S, Lannoy M, Behmoaras J, Pezet M, Sloboda N, Lacolley P, Escoubet B, Buján J, Jacob MP. Potassium channel openers increase aortic elastic fiber formation and reverse the genetically determined elastin deficit in the BN rat. Hypertension. 62:794-801 (2013).